# PrIntMesh: Precise Intersection Surfaces for 3D Organ Mesh Reconstruction

## Abstract

Human organs are composed of interconnected substructures whose geometry and spatial relationships constrain one another. Yet, most deep-learning approaches treat these parts independently, producing anatomically implausible reconstructions. We introduce PrInt-Mesh, a template-based, topology-preserving framework that reconstructs organs as unified systems. Starting from a connected template, PrIntMesh jointly deforms all substructures to match patient-specific anatomy, while explicitly preserving internal boundaries and enforcing smooth, artifact-free surfaces. We demonstrate its effectiveness on the heart, hippocampus, and lungs, achieving high geometric accuracy, correct topology, and robust performance even with limited or noisy training data. Compared to voxel- and surface-based methods, PrInt-Mesh better reconstructs shared interfaces, maintains structural consistency, and provides a data-efficient solution suitable for clinical use.

## 1 Introduction

Human organs are not single solids but interconnected systems. The heart, for example, consists of four chambers—2 atria and 2 ventricles—separated by thin walls, linked by valves, and threaded with vessels. These shapes are tightly coupled; the deformations of one impact the others. Modeling the heart thus requires capturing these relationships, not just segmenting individual chambers. Even small misalignments can compromise anatomical plausibility.

Most current deep-learning approaches ignore this. They break organs apart, predicting substructures such as the left atrium or septal wall independently and ignoring their spatial or functional interdependence. This disrupts continuity and limits use in downstream tasks such as biomechanics (Updegrove et al., 2017; Salvador et al., 2024), electrophysiology (Li et al., 2024; Sakata et al., 2025), and surgical planning (Sakata et al., 2024). Voxel-based models like nnU-Net (Isensee et al., 2021) assign labels independently across the volume, often leaving gaps and disconnected fragments, as in Fig. 1a. Surface-based methods (Wickramasinghe et al., 2020; Kong et al., 2021; Kong & Shadden, 2021; Bongratz et al., 2022; Yang et al., 2023) reconstruct each chamber separately, with no guarantee of proper alignment at shared interfaces, as in Fig. 1b. The resulting meshes are often anatomically implausible and clinically unreliable.

To remedy this, we propose PrIntMesh, a method that reconstructs organs as unified, connected systems rather than collections of parts. Instead of attempting to reconcile independently predicted substructures as most earlier approaches do, PrIntMesh starts from a template encoding anatomical connectivity. It acts as a topological scaffold, ensuring structural integrity throughout deformation. Guided by volumetric image features, the network deforms this scaffold to match individual anatomy while preserving its organization.

PrIntMesh comprises two modules, a feature extractor that processes volumetric imagery and a mesh deformation network that deforms the whole connected template. Unlike earlier models that predict each component independently (Wickramasinghe et al., 2020; Kong et al., 2021), it deforms all components together, preserving spatial relationships by construction. Anatomical realism is maintained by explicitly supervising pairwise interfaces between substructures to preserve coherent internal walls, and by applying geometric regularization terms that encourage smooth, artifact-free surfaces.

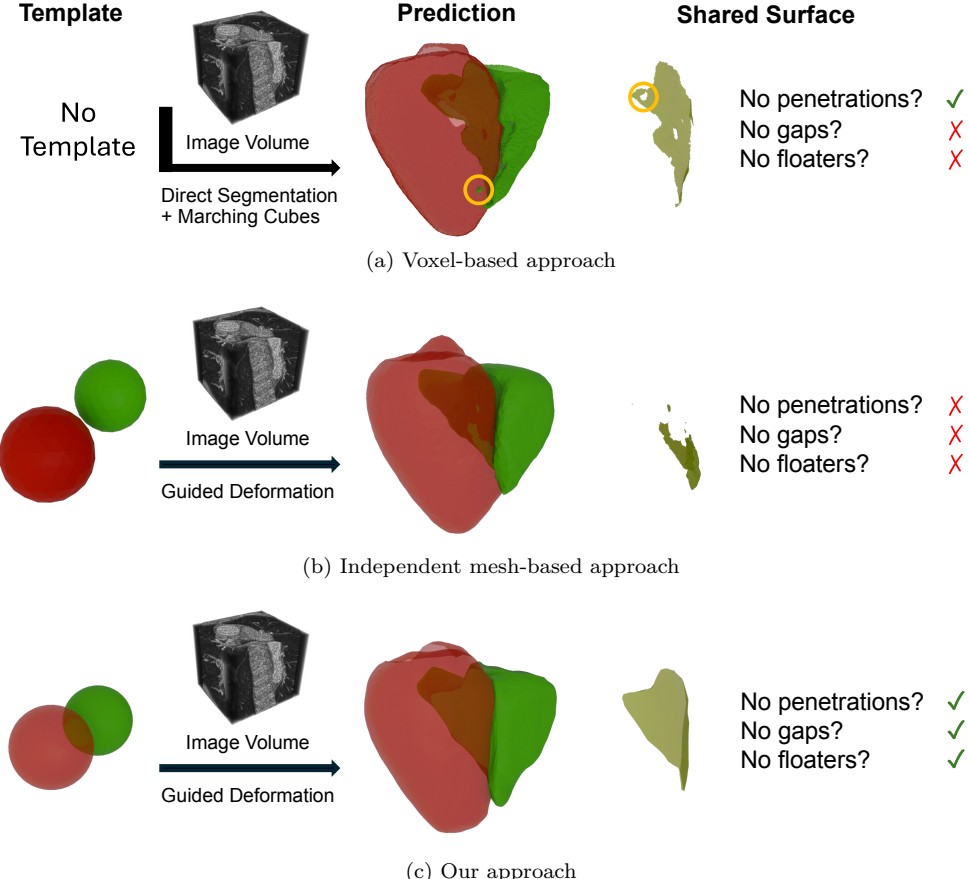

Figure 1: **Voxel-based vs Mesh-based approaches.** (a) Voxel-based approaches such as nnU-Net Isensee et al. (2021) currently are the dominant ones for organ segmentation. However, they can yield gaps, floaters, and jagged boundaries, requiring further post-processing. (b) Existing mesh-based methods generate smooth surfaces but model each class separately, leading to interpenetrations and misaligned boundaries between components. (c) PrIntMesh deforms a joint template for all components and precisely reconstructs shared surfaces, producing a smooth, watertight and topologically correct mesh, without post-processing.

On high-resolution cardiac data, PrIntMesh achieves both high geometric accuracy and guaranteed topological consistency—a combination unmatched by competing methods. While voxel models may yield similar voxelwise scores, they fail to maintain connectivity or realistic internal boundaries. We further tested PrIntMesh on hippocampus and lung datasets, demonstrating generality across anatomies with distinct topologies—from the folded hippocampus to disconnected lungs with imperfect annotations. In all cases, PrIntMesh produces structurally valid, smooth reconstructions even with limited training data.

Crucially, the structural prior embedded in the template enables data-efficient learning. For lungs, PrIntMesh yields a 2–3× lower Chamfer error than nnU-Net (Isensee et al., 2021) while guaranteeing topological correctness and requiring fewer than 100 samples—highlighting its practicality for clinical deployment, with scarce annotated data and rapid adaptation required.

In short, our contributions are as follows:

- We propose a template-based, topology-preserving framework that reconstructs organs as unified, connected systems rather than independent parts.

- We use our templates to strictly enforce anatomical interfaces and geometric regularity, ensuring smooth, artifact-free surfaces and correct internal boundaries.

- We demonstrate high-accuracy, data-efficient reconstruction across diverse anatomies—including heart, hippocampus, and lungs—even given limited or imperfect training data.

The code will be made publicly available.

## 2 Related Work

Deep learning has become the de facto standard for medical image segmentation, including segmenting the liver (Zhang et al., 2023), the whole heart (Yang et al., 2023), and brain tumors (Zeng et al., 2023; She et al., 2023). Most methods output voxel-wise labels using volumetric networks such as nnU-Net (Isensee et al., 2021) or atlas-based registration approaches (Bai et al., 2013; Yang et al., 2018; Iglesias & Sabuncu, 2015). These representations assign a single label per voxel, making it impossible to directly model shared surfaces or multi-part structures. While this may not severely affect standard metrics like Dice, it often leads to topological errors requiring post-processing.

Topology, however, is crucial for downstream modeling. Hemodynamic simulations demand watertight, connected geometries for stable CFD (Updegrove et al., 2017); biomechanics modeling relies on anatomically continuous walls to predict stresses (Salvador et al., 2024); electrophysiological models require consistent atrial and ventricular meshes for accurate excitation propagation (Li et al., 2024; Sakata et al., 2025); and surgical planning or disease progression studies depend on coherent 3D anatomy (Sakata et al., 2024; Pak et al., 2024). As large-scale digital twins emerge (Qian et al., 2025; Qiao et al., 2025), manual mesh repair usually involving voxel-to-surface conversion followed by manual correction (Fischl, 2012; Charton et al., 2021), becomes infeasible.

To explicitly address topological errors, some approaches incorporate topological constraints directly into the learning process (Yang et al., 2023; Hu et al., 2019; Xu et al., 2025; Gupta et al., 2022). Early methods leveraging persistent homology (Hu et al., 2019) compute topological invariants—such as Betti numbers—to formulate penalty losses, which often introduces substantial computational overhead during training. More recent algorithms, such as convolutional interaction modules (Gupta et al., 2022), reduce the computational burden by penalizing incorrect containment and exclusion efficiently. However, because all these methods rely on soft loss formulations applied during training rather than explicit structural priors, they provide no hard guarantees against local artifacts or interpenetrations at inference time.

Implicit representations based on Signed Distance Fields (SDFs) (Yang et al., 2024; Verhülsdonk et al., 2024; Le et al., 2023) and neural rendering (You et al., 2023) offer an alternative to voxels by learning latent shape priors or continuous boundary fields to match voxel segmentations. While these continuous formulations address the discretization artifacts of voxel grids by yielding sub-voxel boundary precision, they require topologically correct training data or soft losses and also lack hard topology guarantees. Furthermore, latent SDF methods tend to be computationally expensive due to their reliance on test time iterative refinement.

Another direction is to directly predict triangulated meshes (Wickramasinghe et al., 2020; Bongratz et al., 2022; Kong & Shadden, 2021; Yang et al., 2023) when internal structures are irrelevant, which yields accurate boundaries. When they are important as in the heart, separate templates are often used per chamber (Wickramasinghe et al., 2020; Kong et al., 2021; Kong & Shadden, 2021; Bongratz et al., 2022; Yang et al., 2023), sometimes with per-template scaling or translation. Independent deformation of these templates, however, causes interpenetrations and gaps that are difficult to fix post hoc.

In contrast, our approach deforms a single unified template encoding internal structures, enabling topologically consistent multi-component reconstructions in a single forward pass without post-processing.

## 3 Method

Our method reconstructs anatomically consistent multi-part organ surfaces from volumetric medical images using a single, unified mesh template. This template encodes the correct topology of the organ and is deformed using learned image features to fit a specific patient's anatomy. The key components of our approach are:

1. A **topologically-correct mesh template** that models all substructures and their interfaces jointly.

2. A **feature-guided mesh deformation network** that aligns and adapts this template to medical imaging data.

3. A **training strategy with combined loss functions** that encourages both overall shape accuracy and precise inter-substructure surface modeling.

We describe each one below, using heart reconstruction for illustration purposes. The method applies similarly hippocampus and lung reconstruction by simply changing the template we deform.

## 3.1 Building a Topologically Correct Template

Our algorithm starts with a template that captures the topological structure of the target organ. For the heart, this includes the four chambers—left/right ventricles and atria—connected by biologically plausible shared walls, such as the myocardium and septa. Our approach to designing the template is to start from a basic primitive geometry that naturally supports defining interconnected components.

To this end, we use a rhombicuboctahedron (Kepler, 1619), illustrated in Fig. 2a. It has 26 faces, consisting of 8 equilateral triangles and 18 squares, which makes it a good approximation of a sphere while also being easy to connect to other rhombicuboctahedra by having them share their square faces. We exploit this property by positioning four rhombicuboctahedra that share faces as shown in Fig. 2b. We then subdivide all faces into smaller facets labeled with a class label denoting one or more substructures. Finally we adjust the position of individual vertices to make the four chambers initially spherical. This yields the initial template of Fig. 2c, which reflects the anatomical layout, that is, adjoining chambers, within a single, connected mesh.

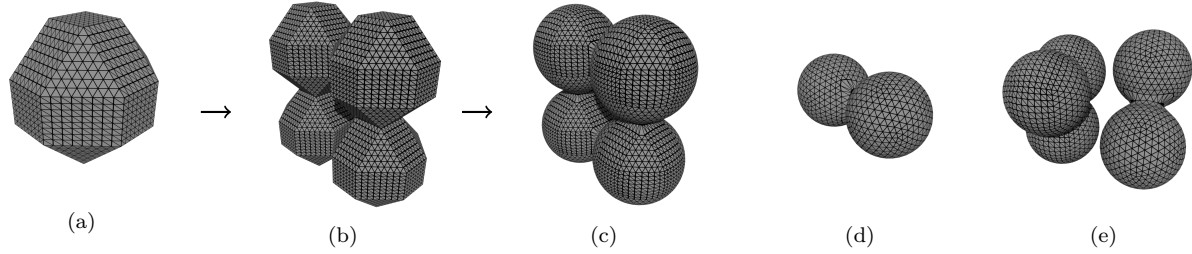

Figure 2: **Building the heart template.** (a) Initial rhombicuboctahedron. (b) Four rhombicuboctahedra glued together. (c) Their vertices are uniformly displaced to create four spheres and complete the template. Templates for (d) the hippocampus and (e) the lungs are constructed similarly.

For the hippocampus and its two parts, we built the template of Fig. 2d, using only two spheres sharing one intersection instead of four. For the lung template of Fig. 2e, we used two spheres for the two lobes of the right lung and three connected ones for the three lobes of the left lung.

While the manual assembly of these primitives is a one-time, offline process that typically accounts for less than 1% of the total training time, we recognize that different anatomical structures may benefit from a more hands-off approach. To address this, we also provide an automated template generation scheme that derives the mesh directly from data. We run the marching cubes algorithm (Lorensen & Cline, 1987) on a representative sample and fuse triangles from different classes that appear in the same location into shared interfaces. Optionally, we apply 3D morphological operations to close holes in the sample before running marching cubes. We then smooth and decimate the final template mesh to reduce computational load. The full procedure is described in Appendix C.1. It ensures that the template inherits the correct topology without requiring manual primitive placement. The experiments described in Appendix C.1 demonstrate that using templates generated in this manner yields a raw Chamfer distance performance nearly identical to using our manually constructed versions, albeit with final triangulations that are more irregular. When regularity is not required by the downstream applications, the automatically generated templates are therefore a viable alternative to the manually constructed ones.

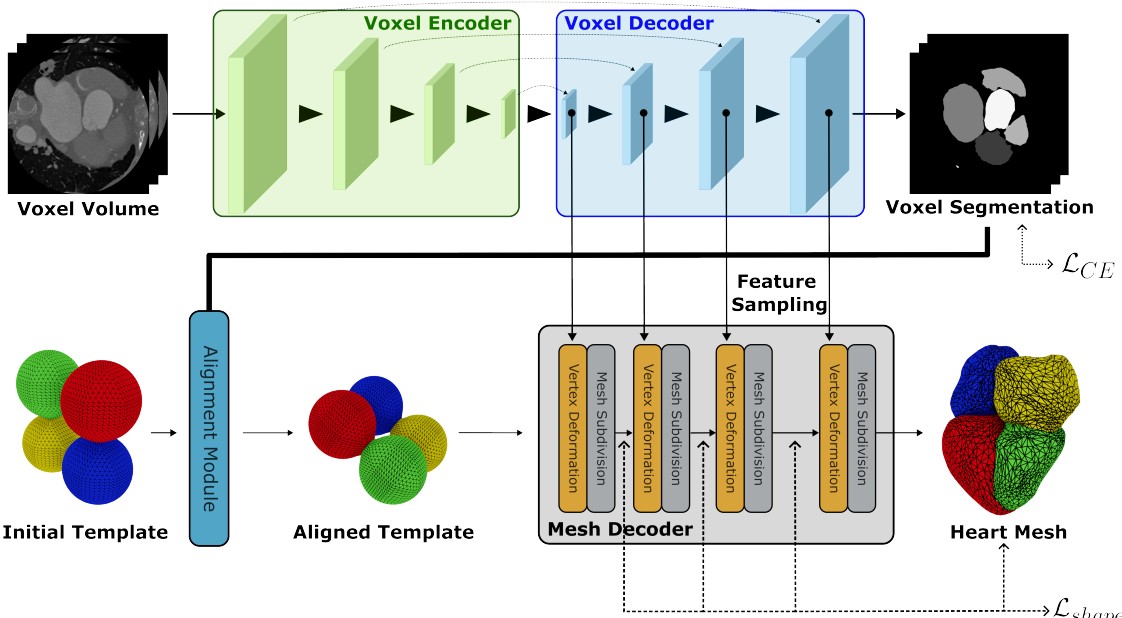

Figure 3: **Network Architecture.** The voxel encoder and decoder follow a U-Net architecture including skip-connections, and are trained with cross-entropy loss. The initial heart template goes through a simple alignment procedure based on the segmentation, and is then deformed and subdivided in multiple steps by the mesh decoder. The mesh decoder samples features from the voxel decoder to apply correct deformations, and the shape loss is applied to each intermediate mesh.

## 3.2   Deforming the Template to Match the Data

A correct, unified template is our starting point. A core challenge then lies in deforming it to accurately reconstruct the anatomy of a specific patient from volumetric medical images, while preserving both topology and mesh regularity as described below. To this end, we use the two-stream network architecture of Fig. 3 to compute translation vectors for each template vertex by combining voxel-based semantic understanding with mesh-based surface generation, as in Voxel2Mesh (Wickramasinghe et al., 2020).

### 3.2.1   Network Architecture

The first stream is a 3D U-Net (Ronneberger et al., 2015) that outputs a dense voxel-wise segmentation and multi-scale feature maps. These maps are fed to the second stream to guide the mesh deformation and to compute an affine transformation that coarsely aligns the template to the target anatomy. Because our template is not radially symmetric, this alignment is critical to avoid unnecessary deformation. We estimate the transform from the predicted class centroids of the segmentation; the computation is fast, differentiable, and performed at every training iteration.

The second stream is a mesh decoder that deforms the aligned template in a coarse-to-fine fashion. At each stage, features are sampled from the voxel stream and used to update vertex positions. Instead of learning curvature-aware unpooling as in (Wickramasinghe et al., 2020), we adopt a simpler subdivision scheme with specific layers quadrupling the number of triangles by splitting each face uniformly. This allows the mesh to gradually increase resolution while preserving class labels on vertices and faces, including those on shared interfaces.

To implement both streams, we started from the publicly available Voxel2Mesh code (Wickramasinghe et al., 2020) and modernized it by replacing the hand-designed adjacency matrix based graph convolutions by newer, more efficient *PyTorch 3D* versions (Ravi et al., 2020). This greatly reduced memory usage and running time, allowing us to work with larger input image resolutions and higher output vertex counts.

### 3.2.2 Preserving Topology via Interface Decomposition

In standard frameworks, each deformable surface is encouraged to match its own target shape based on local evidence. In contrast, in ours, the shared interfaces of our unified template, such as the septal walls between heart chambers, can be pulled in conflicting directions, leading to drift or mesh collapse and, thus, incorrect results as we demonstrate experimentally in Section 4.5. To prevent this, we introduce a topological decomposition of the template and ground truth into disjoint functional regions. Formally, given a set of base anatomical classes $\mathcal{C}_{base} = \{c_1, \ldots, c_k\}$, we define a new supervision set $\mathcal{S}$ comprising:

1. **Exclusive Surfaces:** $S_i = c_i \setminus \bigcup_{j \neq i} c_j$, representing parts of an organ not in contact with others.

2. **Interface Surfaces:** $S_{i,j} = c_i \cap c_j$, representing the precise manifold where two structures meet.

This changes the learning objective from matching $K$ overlapping volumes to matching $|\mathcal{S}|$ non-overlapping components. To ensure stable convergence, we apply this supervision after each stage $l$ of deformation in the graph convolutional network by minimizing

$$\mathcal{L}_{match}^{(l)} = \sum_{s \in \mathcal{S}} \lambda_s \mathcal{L}_{\text{Chamfer}}(P_s^{(l)}, G_s) \,, \tag{1}$$

where $P_s^{(l)}$ represents the points sampled from region $s$ at stage $l$, and $G_s$ are the corresponding ground-truth points. These points are sampled randomly from the underlying meshes, usually 16K points. By assigning equal weight $\lambda_s$ to each region, we ensure that shared interfaces provide a strong gradient signal throughout the refinement process, preventing them from being dwarfed by the larger primary structures.

### 3.2.3 Preserving Mesh Regularity

In addition to preserving topology, geometric integrity must also be maintained. Without it, one is likely to observe vertex collapse, irregular triangle areas, and tangled regions, especially near points of contact between substructures. This would not only degrade reconstruction quality but also render the mesh unsuitable for downstream tasks such as simulation or quantitative analysis. Critically, once such degradations occur, they are difficult or impossible to repair in post-processing. To eliminate such artifacts, we introduce a geometric regularization loss $\mathcal{L}_{\text{reg}}$ that, when minimized, forces the preservation of mesh quality by ensuring overall smoothness, avoiding vertex collapse, and promoting a uniform triangle structure. This eliminates the need for post hoc alignment or mesh repair steps, which are typically error-prone and inconsistent. We take this loss to be

$$\mathcal{L}_{\text{reg}}^{(l)} = \lambda_1 \mathcal{L}_{\text{Edge}}^{(l)} + \lambda_2 \mathcal{L}_{\text{EdgeUnif}}^{(l)} + \lambda_4 \mathcal{L}_{\text{Norm}}^{(l)} + \lambda_5 \mathcal{L}_{\text{Lapl}}^{(l)} \,, \tag{2}$$

where $l$ again denotes the stage of the mesh decoder, the $\lambda$ coefficients are scalar weights, and $\mathcal{L}_{\text{Edge}}^{(l)}$, , $\mathcal{L}_{\text{Norm}}^{(l)}$, and $\mathcal{L}_{\text{Lapl}}^{(l)}$ are standard regularization terms that we define in Appendix A. $\mathcal{L}_{\text{EdgeUnif}}^{(l)}$ is specific to our approach and penalizes edge lengths deviations to keep the triangles uniform.

$$\mathcal{L}_{\text{EdgeUnif}}^{(l)} = \sqrt{\frac{1}{E-1} \sum_{i=1}^{E} (|e_i| - \mathcal{L}_{\text{Edge}}^{(l)})^2} \,,$$

Because our component submeshes are open surfaces and do not lend themselves well to smoothing by regularizers designed for closed surfaces, we compute these losses on the full mesh, rather than for each class separately which is the case for $\mathcal{L}_{\text{Chamfer}}$ in Section 3.2.2. Shared vertices receive contributions from neighbors in different sub-meshes, with appropriate weights.

### 3.2.4 Global Loss

In the previous sections we introduced $\mathcal{L}_{\text{reg}}^{(l)}$ and $\mathcal{L}_{\text{match}}^{(l)}$ whose minimization promotes mesh regularity and topology preservation. To encourage an accurate fit to the training annotations, we also define $\mathcal{L}_{CE}$, the cross-entropy loss, and $\mathcal{L}_{Dice}$ is the Dice loss (Falk et al., 2018) with respect to the ground truth, both computed voxel-wise. When training the network, we minimize the global loss

$$\mathcal{L} = \mathcal{L}_{CE} + \mathcal{L}_{Dice} + \frac{1}{L} \sum_{l=1}^{L} \left( \lambda_{\text{match}} \mathcal{L}_{\text{match}}^{(l)} + \lambda_{\text{reg}} \mathcal{L}_{\text{reg}}^{(l)} \right). \tag{3}$$

## 4  Experiments

We use three datasets to validate our approach on heart, lung, and hippocampus reconstruction, all of which incorporate internal structures that need to be modeled accurately. The templates we use for heart, hippocampus and lung reconstruction have been introduced in Section 3.1 and are depicted by Fig. 2. Further preprocessing and architecture details are provided in Appendix B, along with additional ablation studies in Appendix C.

### 4.1  Datasets

**MM-WHS Heart Dataset** (Zhuang, 2018). It contains 20 whole-heart annotated CT scans for training and 40 CT scans, without publicly available annotations. Seven classes are defined: the left ventricle blood cavity (LV-), the right ventricle blood cavity (RV), the left atrium blood cavity (LA), the right atrium blood cavity (RA), the myocardium of the left ventricle (Myo), the ascending aorta (Ao), and the pulmonary artery (PA). In our experiments, we only model the four heart chambers. Thus, we combine left ventricle blood cavity (LV-) and its myocardium (Myo) into a single left ventricle class (LV), and ignore the Ao and PA classes. We also define three *intersection surfaces* that correspond to the interface surfaces shared by adjacent substructures. Specifically we consider LV ∩ RV, LV ∩ LA, and RV ∩ RA but not LA ∩ RA because the atrial septum is not consistently labeled in the dataset due to the annotations being intended for blood cavities rather than the complete atrial structure. This yields a total of seven classes that we explicitly model.

Without test set annotations, we cannot directly compute metrics on it. Instead, we perform 5-fold cross-validation on the training set, splitting into 5 sets of 16 training and 4 validation samples. We train a separate model for each split so that every one of the 20 annotated scans is held out exactly once, and report the mean and standard deviation over these 20 pooled held-out predictions.

**MSD Hippocampus Dataset** (Antonelli et al., 2022). It comes from the Medical Segmentation Decathlon and comprises 260 small voxel volumes from MRI scans, with the hippocampus split into its anterior and posterior components. We define Anterior ∩ Posterior as the only interface class, which yields a total of three distinct classes. We split the dataset into 200 training samples and 60 test ones.

**TS Lung Dataset** (Wasserthal et al., 2023). For lung reconstruction, we use images and annotations from the TotalSegmentator data. It comprises 1228 CT scans annotated and refined using an active learning setup with a human-in-the-loop. We extract those having viable lung annotations, yielding 348 annotated lung scans. Five classes are defined, the lower, middle and upper right lobe, which we designate as LR, MR and UR, and the lower and upper left lobe, designated as LL and UL. We also consider the four interface classes LR ∩ MR, LR ∩ UR, MR ∩ UR and LL ∩ UL, yielding nine separate classes.

### 4.2  Baselines

We benchmark our approach against four baselines that were selected to represent state-of-the-art versions of the dominant paradigms in 3D medical shape modeling that rely on volumetric, explicit surface mesh, implicit continuous, and hybrid representations.

**nnU-Net** (Isensee et al., 2021; 2024). Representing the gold standard in volumetric segmentation, we utilize the recent residual architecture. While inherently immune to self-intersections, voxel-wise predictions lack

global shape awareness. This frequently yields disconnected components, often referred to as *floaters*, along with jagged boundaries and potential gaps when resolving multi-organ interfaces.

**MeshDeformNet** (Kong et al., 2021). Serving as our explicit mesh baseline, this multi-class extension of Voxel2Mesh (Wickramasinghe et al., 2020) initializes class-specific templates via learned translation and scaling. Though robust against volumetric floaters, the deformation process lacks spatial collision constraints, making it highly susceptible to erroneous inter-class intersections.

**DeepCSR** (Cruz et al., 2021). To evaluate implicit continuous representations, we include this pure Signed Distance Function (SDF) model. DeepCSR maps coordinates to SDF values, extracting the final surface via Marching Cubes. While providing a smooth continuous domain, the independent per-class SDF predictions tend to result in overlapping boundaries.

**MedTet** (Chen et al., 2024). Representing modern hybrid approaches, this framework utilizes Deep Marching Tetrahedra (Shen et al., 2021). By combining a deformable tetrahedral grid with point-wise SDF predictions, it achieves high reconstruction accuracy. However, because it processes structures independently, it still cannot guarantee intersection-free boundaries in dense multi-part configurations.

Some of these implementations are now a few years old but not yet been superseded by newer methods. We provide more implementation details in Appendix B.3.

## 4.3 Evaluation Metrics

The primary goal of our approach is to produce anatomically faithful 3D reconstructions with correct topology—maintaining the intended connectivity between structures, avoiding interpenetration, and preserving interface surfaces. These structural properties are essential for both clinical relevance and downstream applications such as simulation. However, they are not adequately captured by standard segmentation metrics, which typically focus on voxel-wise or surface-level similarity. To fully assess reconstruction quality, we report both **topology-aware metrics**, which directly evaluate structural correctness, and **standard metrics**, which provide complementary assessments of overall volume alignment. This dual evaluation highlights the limitations of conventional metrics in avoiding anatomical errors and underscores the strengths of our method in preserving topological integrity. We describe both kinds below.

### 4.3.1 Topology Metrics

Topological correctness is difficult to assess using standard metrics. Structural errors such as unintended overlaps or missing connections often have little effect on global scores like Dice or Chamfer, which average over entire volumes or surfaces. Localized shifts at boundaries may only impact a small number of voxels, but these small inconsistencies can lead to artifacts that compromise anatomical validity and limit the utility of the reconstruction in downstream tasks.

Such topological errors arise in different ways depending on the representation. Mesh-based methods may generate interpenetrating surfaces between structures that should only be adjacent, while voxel-based methods—particularly those that segment each structure independently—can leave small gaps or disconnected regions where continuity is required. Because enforcing topological correctness has not been a primary focus in the field of organ reconstruction, there are no standard metrics to quantify topological mistakes. Thus, we designed two such metrics to quantify structural inconsistencies at interfaces: one measuring the volume of mesh intersections, and another capturing the presence of undesired gaps between components.

**Volume Intersections.** For mesh-based methods that generate erroneous intersections, we use mesh boolean operations to extract the intersection between meshes corresponding to different classes and compute its volume relative to the total mesh volume.

**Unwanted Gaps.** Voxel-based methods can produce unintended empty regions inside the heart due to disconnected or misaligned surfaces. We quantify these artifacts by counting the number of isolated background (empty) connected components that are entirely enclosed within the structure—excluding the outer background. In this context, each such enclosed component corresponds to a "hole" or gap.

Table 1: Per-class Chamfer distance ($\times 10^{-3}$) and normal consistency on MM-WHS-4. *Blue denotes the best method and light blue the second best. Our method is consistently one of the two and, when it is second-best, it is only by a small margin.*

| | Interface Surfaces | | | | | | | | | | | | Base Classes | |
| | LV ∩ LA | | | LV ∩ RV | | | RV ∩ RA | | | Average | | | Average | |
| Method | CD ↓ | NC ↑ | VIR ↑ | CD ↓ | NC ↑ | VIR ↑ | CD ↓ | NC ↑ | VIR ↑ | CD ↓ | NC ↑ | VIR ↑ | CD ↓ | NC ↑ |
|---|---|---|---|---|---|---|---|---|---|---|---|---|---|---|
| Ours | $0.7 \pm 0.5$ | $0.95 \pm 0.02$ | 100% | $1.6 \pm 1.3$ | $0.95 \pm 0.01$ | 100% | $1.6 \pm 2.8$ | $0.95 \pm 0.02$ | 100% | $1.3 \pm 1.9$ | $0.95 \pm 0.02$ | 100% | $1.0 \pm 0.9$ | $0.93 \pm 0.03$ |
| MedTet | $3.6 \pm 5.5$ | $0.93 \pm 0.02$ | 95% | $3.2 \pm 3.2$ | $0.96 \pm 0.01$ | 100% | $2.6 \pm 4.2$ | $0.95 \pm 0.02$ | 100% | $3.1 \pm 4.3$ | $0.94 \pm 0.02$ | 98% | $1.1 \pm 1.2$ | $0.93 \pm 0.03$ |
| DeepCSR | $3.6 \pm 4.8$ | $0.94 \pm 0.02$ | 80% | $7.4 \pm 9.1$ | $0.95 \pm 0.02$ | 80% | $4.3 \pm 7.7$ | $0.95 \pm 0.02$ | 85% | $5.1 \pm 7.5$ | $0.95 \pm 0.02$ | 82% | $1.1 \pm 1.3$ | $0.94 \pm 0.03$ |
| MeshDeformNet | $2.8 \pm 4.4$ | $0.91 \pm 0.05$ | 100% | $6.1 \pm 11.$ | $0.89 \pm 0.08$ | 100% | $8.5 \pm 16.$ | $0.89 \pm 0.06$ | 100% | $5.8 \pm 12.$ | $0.89 \pm 0.07$ | 100% | $2.5 \pm 3.2$ | $0.88 \pm 0.05$ |
| nnU-Net | $1.0 \pm 0.8$ | $0.88 \pm 0.02$ | 100% | $1.8 \pm 1.6$ | $0.89 \pm 0.02$ | 100% | $1.9 \pm 2.7$ | $0.90 \pm 0.02$ | 100% | $1.6 \pm 1.9$ | $0.89 \pm 0.02$ | 100% | $1.1 \pm 1.0$ | $0.87 \pm 0.03$ |

Table 2: Comparing Dice coefficients. *Our meshes are watertight. After voxelization, which reduces precision, their volumetric accuracy is similar to nnU-Net.*

| | MM-WHS | | | | | MSD-Hippocampus | | |
| Method | LV | RV | LA | RA | Mean | Anterior | Posterior | Mean |
|---|---|---|---|---|---|---|---|---|
| Ours (Mesh) | **0.95** | **0.90** | **0.92** | 0.88 | 0.91 | 0.87 | 0.85 | 0.86 |
| nnU-Net | **0.95** | **0.90** | **0.92** | **0.89** | **0.92** | **0.88** | **0.86** | **0.87** |

Table 3: Topology metrics on MM-WHS-4.

| Method | Intersection Volume | # of Unwanted Gaps |
|---|---|---|
| Ours | **0** | **0** |
| nnU-Net | **0** | $0.85 \pm 0.02$ |
| MedTet | $0.18\% \pm 0.08\%$ | undefined |
| DeepCSR | $0.08\% \pm 0.09\%$ | undefined |
| MeshDeformNet | degenerate meshes | undefined |

### 4.3.2 Standard Metrics

In addition to topology metrics, we report standard evaluation metrics commonly used in segmentation benchmarks. While these do not directly assess structural integrity, they offer a baseline for geometric similarity. They come in two main flavors, volumetric and surface-based. Thus we consider the following three.

**Chamfer Distance.** It is calculated between two point clouds. For the mesh-based methods, we obtain them by sampling 1024K points from the output surfaces. For the volumetric ground-truth data and the output of nnU-Net, we get them by running the Marching Cube algorithm (Lorensen & Cline, 1987) and then sampling. We calculate it for every class, including interfaces between substructures. All calculations are done with vertex coordinates normalized into the [-1, 1] range, and we report values multiplied by $10^{-3}$.

**Normal Consistency** (Gkioxari et al., 2019). It is calculated in a very similar way to Chamfer distance by comparing the orientation of surface normals at matching points rather than their distance. Based on the cosine distance, its best value is 1 when the normals match perfectly, and 0 when they are completely orthogonal. Normals at random points on the mesh surface are obtained by interpolating vertex normals.

**Dice Coefficient** (Dice, 1945). It measures volumetric overlap between ground truth and prediction, while being more sensitive to class imbalance than the Jaccard index (Bertels et al., 2019). We compute it for each segmentation class. For mesh based methods such as ours, we first convert the mesh representation into a voxel-based one to compute this metric.

### 4.4 Comparative Results

We now provide the performance of PrIntMesh against that of our voxel- and surface-based baselines and argue that our method effectively combines the strengths of both: Voxel-based methods often succeed in reconstructing intersections due to contact, but the resulting surfaces tend to be much rougher. They can also produce floaters. Conversely, surface-based reconstructions tend to produce smoother surfaces but can easily fail in producing a sensible intersection at contact areas. PrIntMesh addresses all these issues jointly.

In the tables below we report our metrics separately for the exclusive and interface surfaces, that is, regions where anatomical structures meet as opposed to regions belonging to a single structure, as defined in Section 3.2.2. Since interfaces are not explicitly modeled by MeshDeformNet, MedTet, and DeepCSR, we use boolean operations to extract approximate ones from the meshes to compute our metrics. Because not all samples produce valid interfaces—some intersection operations may yield an empty set—we also report the percentage of samples that do. When calculating Chamfer distance and normal consistency for interface

Table 4: Paired, two-sided Wilcoxon signed-rank $p$-values comparing PrIntMesh against each baseline on per-class Chamfer distance, over all 20 held-out hearts (5 folds $\times$ 4 hearts) from MMWHS-4. *Bold marks $p < 0.05$; values below $0.001$ are reported as $< 0.001$.*

| Baseline | Interface Surfaces | | | | Base Classes |
|---|---|---|---|---|---|
| | LV $\cap$ LA | LV $\cap$ RV | RV $\cap$ RA | Average | Average |
| MedTet | **0.040** | **0.017** | 0.430 | 0.064 | 0.869 |
| DeepCSR | **< 0.001** | **0.001** | 0.159 | **< 0.001** | 0.812 |
| MeshDeformNet | 0.064 | **0.003** | 0.070 | **0.004** | **0.006** |
| nnU-Net | 0.114 | 0.596 | 0.261 | 0.216 | 0.927 |

surfaces, we average only over samples for which *all* baselines yielded interfaces for that particular class to ensure the comparisons are done on the same distribution. This represents between 30% and 100% of the samples depending on the dataset and surface. We report it as VIR (Valid Intersection Rate) in the tables.

**Heart Reconstruction.** We report metrics on the MM-WHS-4 dataset in Tab. 1. They are computed on both interface surfaces where anatomical structures meet, as well as the base classes which are the actual heart chambers. When calculating metrics for the base classes with our method, we reassemble them from the separated components, that is, the left ventricle will be the union of the exclusive left ventricle surface, the interface between the left and right ventricles, and the interface between the left ventricle and left atrium. Lower CD and higher NC indicate better surface alignment and smoother geometry.

Our method achieves the best overall performance across the majority of intersection classes, yielding the lowest average Chamfer Distance and highest or near-highest Normal Consistency. Notably, it ranks first in all 5 CD columns and achieves excellent NC performance. This indicates that it not only captures accurate surfaces but also ensures smooth transitions in geometrically complex areas, which is critical when modeling realistic cardiac interfaces. In terms of volume reconstruction accuracy as reported in Tab. 2, our method performs comparably to nnU-Net, which is unsurprising because the differences between reconstructions occur mostly at the boundaries between substructures, which only represent a small fraction of the total number of voxels. Furthermore, its voxel-based nature results in visible artifacts when turning the reconstructed volume into 3D surfaces. As can be seen in the third column of Fig. 4, nnU-Net outputs exhibit rough and uneven walls, especially in the "LV + Walls" and "LV (No Walls)" views. This translates into the comparatively lower NC score of Tab. 1. Such artifacts do not exist in real human anatomy. In reality, organs like the heart have naturally smooth and continuous surfaces, shaped by biological processes and soft tissue mechanics. This problem is not specific to nnU-Net; it is inherent to fixed-resolution voxel-based representations.

Conversely, MedTet and DeepCSR perform significantly worse than us in terms of CD score, especially in challenging regions such as LV-RV and RV-RA, but outperform us slightly in terms of NC scores in the intersection classes, This suggests that while these methods produce smooth normals, they are less good at precisely aligning surfaces.

These observations are further supported by the topological metrics presented in Tab. 3, which quantify structural correctness. Our method achieves zero intersection volume and no unwanted gaps, indicating clean, watertight surface reconstructions. In contrast, nnU-Net, while free from surface self-intersections, introduces an average of nearly one unwanted surface gap per case—further highlighting the challenges of voxel-based reconstruction in topologically complex regions. MedTet, DeepCSR and MeshDeformNet introduce overlapping volumes between structures, and MeshDeformNet frequently outputs degenerate meshes, making a correct volume calculation difficult. The number of gaps is ill-defined for these because they are made of several independent connected components.

In short, by jointly working with surface and volume representations, our method produces smooth, coherent, and anatomically plausible geometry, while closely matching what we expect from actual biological structures. This is especially important in sensitive areas like the LV–RV wall, where precise surface detail matters both visually and functionally.

*Statistical Significance.* Testing for statistical significance on the per-fold results of our cross-validation strongly violates the independence assumptions of a standard $t$-test, due to the training data overlapping

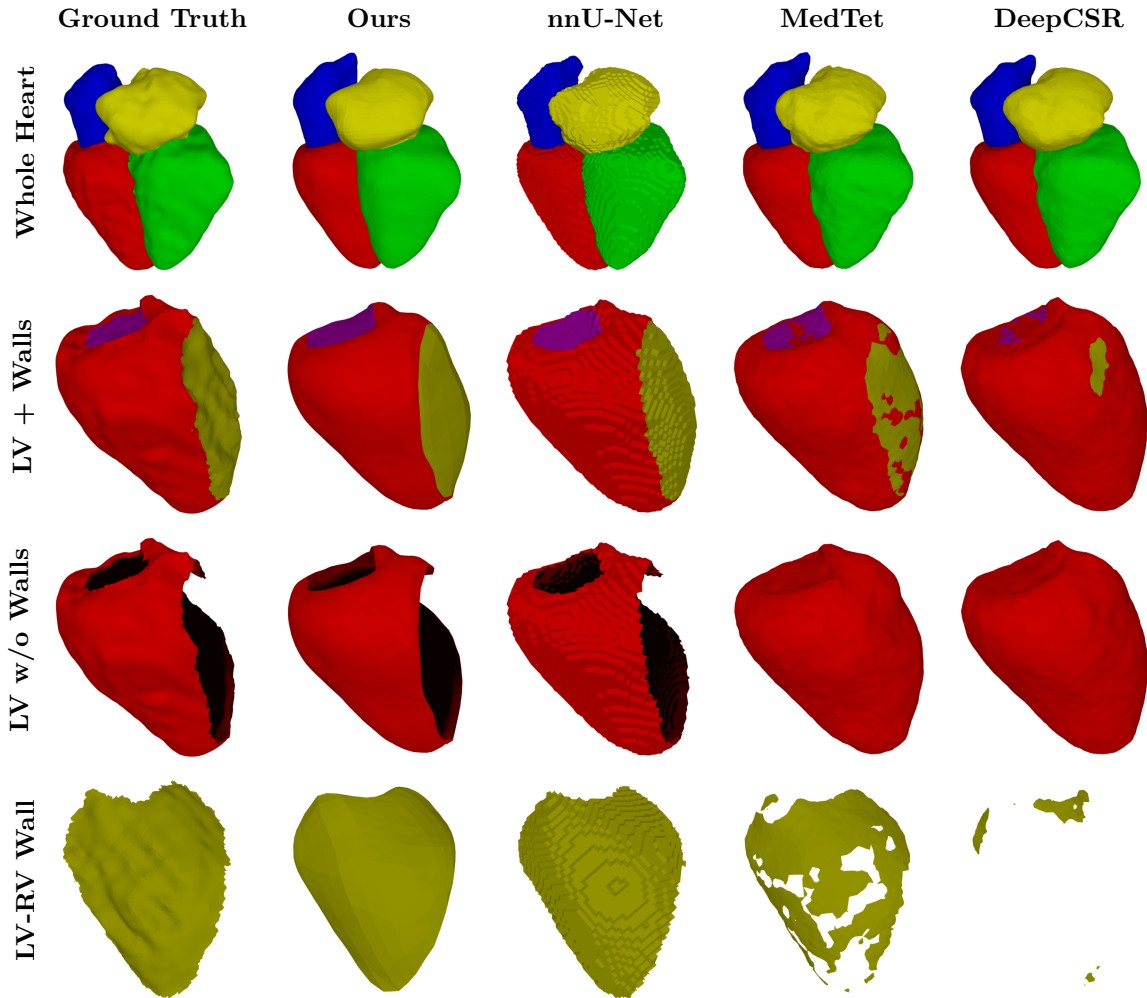

Figure 4: **Qualitative comparison of 4-chamber heart outputs.** Left column describes the anatomical view. Each row compares methods visually. Our method produces smoother and more consistent geometry, especially in edge-sensitive regions.

across folds. We instead pool the out-of-sample predictions from every fold to obtain all 20 held-out hearts, as in Tab. 1, and compare the methods pairwise on each heart. Since the distribution of Chamfer errors is skewed and tends to be long-tailed, and therefore not normally distributed, we compare PrIntMesh against each baseline one by one with a paired Wilcoxon signed-rank test, which avoids the normal distribution assumption of the paired $t$-test. The test is paired on the hearts for which both methods produce the interface. The resulting $p$-values are reported in Tab. 4. They support our conclusion: against most baselines we obtain a clear improvement on the interface surfaces. The only baseline where the difference is far from significance is nnU-Net, which is expected, as its voxel segmentation prevents intersections and is already roughly correct; our comparative advantage is the guarantee of a well-connected and smooth interface surface with no holes or artifacts.

**Hippocampus and Lung Reconstruction.** We report our hippocampus reconstruction results in Tab. 5 with a focus on the junction between anterior and posterior segments, along with qualitative results in Fig. 5. Our method delivers lower CD at this junction with high NC, indicating a smooth and realistic transition. Our method's ability to model this boundary as a clean, topologically consistent surface—without jagged artifacts or mesh discontinuities—demonstrates its strength in capturing anatomically realistic intersections, even in small, structurally delicate regions. We observe similar behavior for lung reconstruction results,

Table 5: Per-class surface metrics on MSD-Hippocampus.

| | Ant. ∩ Post. | | | Anterior | | Posterior | |
|---|---|---|---|---|---|---|---|
| | CD↓ | NC↑ | VIR↑ | CD↓ | NC↑ | CD↓ | NC↑ |
| Ours | **3.2** | **0.98** | **100%** | **1.3** | **0.90** | 1.3 | **0.90** |
| MedTet | 4.0 | 0.97 | 32% | **1.3** | 0.86 | 1.3 | 0.85 |
| DeepCSR | 3.5 | 0.97 | **100%** | **1.3** | 0.86 | 1.3 | 0.85 |
| nnU-Net | 3.5 | **0.98** | **100%** | **1.3** | 0.80 | **1.2** | 0.80 |

Table 6: Per-class surface metrics on TS-Lung.

| | Interface Surfaces | | | | | | | | | | | | | | | Base Classes | |
|---|---|---|---|---|---|---|---|---|---|---|---|---|---|---|---|---|---|
| | LR ∩ MR | | | LR ∩ UR | | | MR ∩ UR | | | LL ∩ UL | | | Average | | | Average | |
| Method | CD↓ | NC↑ | VIR↑ | CD↓ | NC↑ | VIR↑ | CD↓ | NC↑ | VIR↑ | CD↓ | NC↑ | VIR↑ | CD↓ | NC↑ | VIR↑ | CD↓ | NC↑ |
| Ours | **2.61** | **0.98** | **100%** | 0.53 | **0.98** | **100%** | **0.59** | **0.97** | **100%** | 2.52 | **0.94** | **100%** | 1.56 | **0.97** | **100%** | 0.98 | 0.95 |
| MedTet | 3.21 | 0.96 | 65% | 1.02 | 0.97 | 65% | 2.94 | 0.95 | 65% | 2.72 | 0.92 | 72% | 2.47 | 0.95 | 67% | 1.01 | 0.96 |
| DeepCSR | 3.21 | **0.98** | **100%** | 1.72 | **0.98** | **100%** | 0.72 | **0.97** | **100%** | 15.83 | **0.94** | **100%** | 5.37 | **0.97** | **100%** | **0.67** | **0.97** |
| nnU-Net | 2.70 | 0.91 | **100%** | **0.47** | 0.89 | **100%** | 0.65 | 0.91 | **100%** | **1.32** | 0.91 | **100%** | **1.28** | 0.91 | **100%** | 0.69 | 0.89 |

as can be seen in Tab. 6, with superior intersection reconstruction compared to mesh-based methods, and superior normal consistency compared to voxel segmentation along with no topological errors.

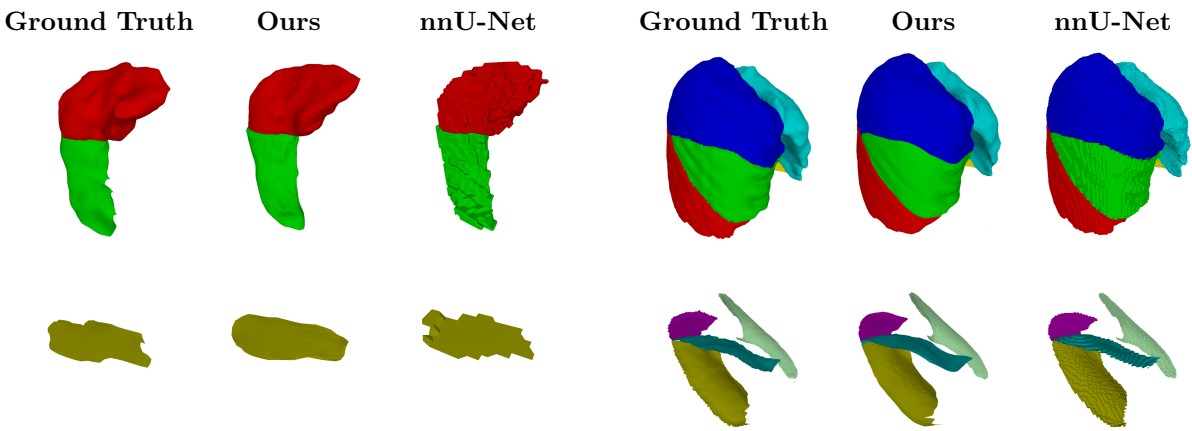

(a) Hippocampus reconstructions and interfaces.  (b) Lung reconstructions and interfaces.

Figure 5: **Qualitative comparisons.** (a) The ground truth hippocampus remains irregular even after smoothing due to the low data resolution. Shared interfaces areas are zoomed in. (b) PrIntMesh yields a smooth lung reconstruction free of residual and topological irregularities and with well-defined intersections. In contrast, the nnU-Net output is much rougher.

### 4.5 Ablation Studies

We study the effect of the two main parts of our design, the explicit supervision of the shared interface surfaces and the template alignment module, on a single split of MMWHS-4. We report further ablations in Appendix C.

**Interface Supervision.** The defining feature of our approach is that we supervise the shared interface surfaces explicitly, rather than letting the adjacent main classes cover them implicitly. To gauge the importance of this, we keep the unified template but supervise only the main classes, so that each interface surface is supervised additively by the two main classes it belongs to. As reported in Tab. 7, this causes a large drop in interface reconstruction quality. The average intersection Chamfer distance increases from 1.22 to 8.23, and from 2.59 to 22.07 on the LV ∩ RV wall. The degradation arises from the ambiguity and loss of identity of the interface surfaces during training. As they are not explicitly supervised, parts of the main classes "leak" to cover the ground-truth interface surface, leaving it only partially reconstructed in the deformed template.

**Template Alignment.** The alignment module applies a similarity transform (rotation, uniform scale, and translation) estimated from the predicted class centroids, bringing the template into the patient's coordinate frame before the deformation network runs. To measure its contribution, we disable this step entirely, so that the deformation network receives the template in its original pose and must recover the rigid displacement and scale on its own. As shown in Tab. 8, the average per-class Chamfer distance rises by 19% on the validation set, with the error concentrated on the intersection classes (+27% vs. +13% for standalone chambers) and inverted facets appearing on several classes. This happens because Chamfer supervision alone is ill-suited

Table 7: The effect of not supervising shared-surfaces in MMWHS-4. *Performance is greatly reduced in terms of Chamfer distance ($\times 10^{-3}$).*

| | Intersection Classes | | | | Other Classes |
|---|---|---|---|---|---|
| Template | LV ∩ LA | LV ∩ RV | RV ∩ RA | Average | Average |
| Shared-surface | 0.31 | 2.59 | 0.75 | 1.22 | 1.05 |
| Main-surface | 0.57 | 22.07 | 2.06 | 8.23 | 1.09 |

Table 8: Effect of disabling the template alignment module on a single split of MMWHS-4. *Chamfer distance ($\times 10^{-3}$) degrades on most classes.*

| | Intersection Classes | | | | Other Classes |
|---|---|---|---|---|---|
| Variant | LV ∩ LA | LV ∩ RV | RV ∩ RA | Average | Average |
| With alignment | 0.37 | 1.75 | 1.00 | 1.04 | 0.98 |
| Without alignment | 0.36 | 2.23 | 1.35 | 1.31 | 1.11 |

Table 9: Training and inference time on MMWHS-4, reported as mean ± standard deviation in seconds.

| Method | Training (s/step) | Inference (s/sample) |
|---|---|---|
| nnU-Net (Isensee et al., 2021; 2024) | $0.182 \pm 0.003$ | $0.330 \pm 0.001$ |
| MeshDeformNet (Kong et al., 2021) | $0.650 \pm 0.010$ | $0.139 \pm 0.004$ |
| DeepCSR (Cruz et al., 2021) | $0.282 \pm 0.035$ | $0.243 \pm 0.001$ |
| MedTet (Chen et al., 2024) | $1.778 \pm 0.155$ | $0.585 \pm 0.037$ |
| **PrIntMesh (Ours)** | $0.510 \pm 0.014$ | $0.113 \pm 0.005$ |

for recovering large global pose and scale offsets, which the alignment module does well. This makes an important contributor to both accuracy and output mesh quality.

### 4.6 Computational Efficiency

We report per-step training time and per-sample inference time for our method and the baselines on MMWHS-4 in Tab. 9. All tests are run on a machine with a V100 GPU and 8 cores of an Intel Xeon Gold 6240 CPU and averaged over 250 steps. The timings depend greatly on implementation details. nnU-Net is the fastest to train because it consists only of a voxel segmentation network, but its inference is the slowest because it performs 8 mirrored predictions and averages them for higher accuracy. MedTet's training is slow because we use a high-resolution grid, which requires predicting and supervising many SDF values in the initial prediction. DeepCSR is much faster during training because it does not perform mesh extraction at training time. Our times are similar to MeshDeformNet's, the most similar method, with the difference coming down to implementation details and small optimizations.

### 4.7 Effect of Training Set Size

In medical applications, large training sets are not always available and being able to train a network using a relatively small dataset is desirable. With this in mind, we re-ran the hippocampus and lung reconstruction experiments, using only a fraction of the training database to train our approach and nnU-Net. We report the results in Fig. 6. In the low-data regime, our approach clearly outperforms nnU-net not only in NC terms but also in CD terms. This is due to the strong prior our approach imposes on the organ shape with the built-in template. While nnU-Net produces artifacts and disconnected regions in the low-data regime and greatly benefits from more data, our approach remains consistent even with few training samples.

## 5 Limitations

Our method assumes a fixed anatomical topology, encoded in the template. For organs with consistent connectivity, this guarantees the intersection-free meshes we reconstruct. It does not extend to anatomies with variable patient-specific connectivity, or to anomalies that break the template's assumptions. The network will still produce an output matching the template's topology in such cases. Multi-template selection or topology-adaptive deformation are natural extensions for future work.

Template alignment depends on the segmentation network, since the similarity transform is estimated from the predicted class centroids. A poor segmentation degrades the alignment and gives the deformation network a bad starting point. Training the two networks jointly reduces this dependency but does not remove it.

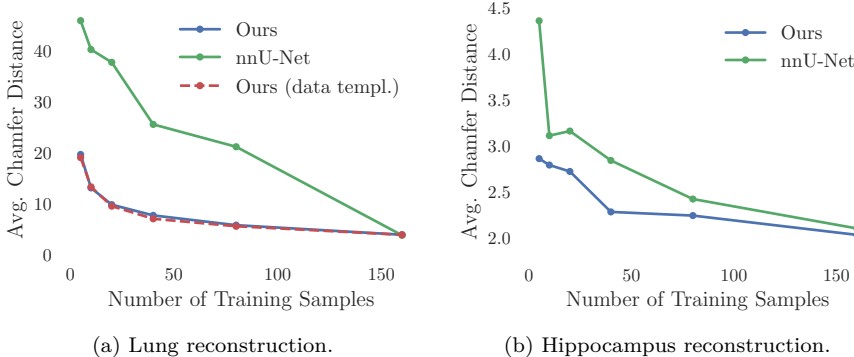

(a) Lung reconstruction.

(b) Hippocampus reconstruction.

Figure 6: **Performance in the low-data regime.** With few training samples available, our approach yields more accurate surface reconstructions than nnU-Net. We also show the reconstruction performance of using a data template for the lung, which is very similar to the performance of using a hand-made template.

Designing a multi-part template requires organ-specific effort. Even when automatically constructing from a sample, it requires careful checking to ensure the topology of the specific sample matches the intent and does not have too many voxelization artifacts.

## 6   Conclusion

We introduced PrIntMesh, a unified framework for jointly reconstructing multiple organ components. Unlike existing surface-based methods that model each component using independent, potentially overlapping meshes, PrIntMesh deforms a single multi-part template that preserves topology while accurately modeling shared interfaces. This enables state-of-the-art reconstruction accuracy with guaranteed topological consistency.

Our approach moves toward fully automatic generation of anatomically correct models suitable for downstream simulations without manual post-processing. Future work will focus on handling more complex anatomies, such as incorporating the aorta and pulmonary artery and capturing their temporal deformations, as well as modeling more detailed interfaces.

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

# A    Geometric Regularization Loss Terms

**Edge Losses $\mathcal{L}_{\mathbf{Edge}}$ and $\mathcal{L}_{\mathbf{EdgeUnif}}$.**   We take them to be

$$\mathcal{L}^l_{\text{Edge}} = \frac{1}{E} \sum_{i=1}^{E} |e_i| \, , \tag{4}$$

$$\mathcal{L}^l_{\text{EdgeUnif}} = \sqrt{\frac{1}{E-1} \sum_{i=1}^{E} (|e_i| - \mathcal{L}^l_{\text{Edge}})^2} \, ,$$

where $E$ is the total number of edges in the mesh and $e_i$ are its edges. Minimizing them discourages the elongation of edges by penalizing both their average length and deviations from this average length, thereby promoting the regularity of the mesh facets.

**Normal Loss $\mathcal{L}_{\mathbf{Norm}}$.** We write it as

$$\mathcal{L}_{\text{Norm}} = \sum_{i=1}^{F} \sum_{j=1}^{F} \text{Adj}(f_i, f_j)(1 - \cos(n_{f_j}, n_{f_i})) \, , \tag{5}$$

$$\text{Adj}(f_i, f_j) = \begin{cases} 1, & \text{if } f_i, f_j \text{ are adjacent faces} \\ 0, & \text{otherwise} \end{cases} \, ,$$

where $F$ is the number of faces in the mesh and $f_i$, $f_j$ are faces. Minimizing it further regularizes pairs of adjacent triangle faces by harmonizing their normals.

**Laplacian Loss $\mathcal{L}_{\mathbf{Lapl}}$.** We take it to be

$$\mathcal{L}_{\text{Lapl}} = \frac{1}{V} \sum_{i=1}^{V} \left\| v_i - \frac{1}{|N_i|} \sum_{j \in N_i} v_j \right\| \, ,$$

where $v_i$ are the vertices, $V$ the number of vertices in the mesh and $N_i$ the set of indices of the vertices neighboring $v_i$. Minimizing it promotes smoothness by pushing each vertex towards the mean of its neighbors (Desbrun et al., 1999).

# B    Architecture and Preprocessing

For our segmentation network, we use nnU-Net in our experiments on the hippocampus and lung, with 5 levels containing [32, 64, 128, 256, 320] channels for the lung and 4 levels for the hippocampus. For our heart experiments, we use a smaller UNet with fewer channels which we find faster to train for comparable results. For the GCN, we use the same structure as in (Wickramasinghe et al., 2020) with a hidden dimension of 32 and as many layers as the segmentation network decoders, and using learned neighborhood sampling (Wickramasinghe et al., 2020). We use the Adam optimizer with a learning rate of $10^{-4}$, $\beta_1 = 0.9$ and $\beta_2 = 0.999$. We set loss multipliers on a per-dataset basis by picking the best from a random search. All experiments are performed using a single V100 GPU.

## B.1    Mesh Data Flow & Decoder Details

**Decoder output.** Each stage of the mesh decoder predicts only a per-vertex 3D deformation, of shape $(V, 3)$ where the input mesh has $V$ vertices. $V$ starts as the number of vertices in the template and grows with each subdivision stage.

**Mesh representation.** The mesh fed to the model has three components. Two are standard in any mesh: a $(V, 3)$ floating-point vertex array storing vertex coordinates, and an $(F, 3)$ integer face array pointing to the three composing vertices of each triangle. We add a third integer array of shape $(F,)$ that stores the class of each triangle. These face classes stay fixed during deformation, with the exception of subdivision stages

where each new triangle inherits the class of its parent. Shared interfaces use a paired encoding: given the two adjoining classes $c_1$ and $c_2$, we encode them as $c = K \cdot c_1 + c_2$ with $K = 1000$. This encoding is only important at evaluation time, when extracting a "full" part; during training, shared interfaces are treated the same as main classes, i.e. individually.

**Subdivision scheme.** We use regular mesh subdivision, where one triangle is divided into four new triangles by adding new edges between the midpoints of each existing edge. We adopted this simple approach rather than the adaptive unpooling of the original Voxel2Mesh (Wickramasinghe et al., 2020) for two reasons. First, caching and re-using the same triangle topology at every training step speeds up training, at the sole cost of having more vertices in the final output and without any significant impact on performance. Second, and perhaps more important, adaptive unpooling has a topological limitation: by design, it first drops a fixed ratio of vertices from the denser regions of the deformed shape, and then retriangulates the original template with a convex hull algorithm. This works when the template is a convex shape such as a simple sphere, as is the case in most previous methods, but not for our unified multi-part templates. If we modified the template in this way, we would not be able to recover a new one via convex hull, and applying a similar approach would require a significant redesign.

**Scale limits.** There is a limit on the number of triangles due to memory, but it is quite high. As an example, our heart experiments use a single subdivision and produce an output mesh with around 43K triangles. In our tests, training fits on a single 80GB A100 GPU with up to 2.7 million output triangles (inference can go even higher). In practice, we find ourselves more limited by compute during training than by memory: at our default size, one iteration takes about 0.4 seconds, while it takes nearly 20 seconds in the 2.7M-triangle setting.

## B.2 Dataset Preprocessing

For the MM-WHS-4 dataset, We preprocess each CT scan by cropping each heart around its bounding box with a 10% extra margin added around the cube for padding. The resulting volume is resized to $128{\times}128{\times}128$ and values are normalized per instance to have zero-mean and unit standard deviation.

For the MSD hippocampus we resize each volume to $64 \times 64 \times 64$ and normalize gray-level values using the mean and standard deviation of the whole training dataset.

For the TS Lung dataset, we resize all volumes to $128 \times 128 \times 128$, and use a random 260/28/60 train/validation/test split for our experiments, with values being normalized per-instance.

## B.3 Baselines

We use the original published code for all baselines and feed the same preprocessed data to each of them.

The only exception is DeepCSR whose original codebase is heavily specialized for brain MRI images. We instead reimplement it in our framework by predicting SDF with a separate MLP for each class. During training, half of the points we sample are uniformly random, and the other half are sampled around the ground truth surface. For the points sampled around the surface, we obtain them by adding Gaussian noise to points sampled from the surface. We apply approximately 1 mm noise to half and 4 mm noise to the other half to have an even mix of coarse and fine supervision.

## C  Ablation Studies

## C.1  Data-Based Template Generation

We use the hand-designed templates of Fig. 2, which we have to re-define for each new organ. Instead, we could derive them directly from labeled ground truth data. To demonstrate this, we triangulated a representative heart segmentation from MM-WHS-4, and decimated the result to generate approximately as many vertices as in our hand-designed template. We then trained our model to use this new template as we did before and evaluate on 4 out of 5 splits, to avoid evaluating on the split the segmentation comes from. In

Table S1: regular template compared to a data template on MMWHS-4. *Results are similar in terms of CD ($\times 10^{-3}$) despite visual differences.*

| Template | Intersection Classes | | | | Other Classes |
|---|---|---|---|---|---|
| | LV ∩ LA | LV ∩ RV | RV ∩ RA | Average | Average |
| Sphere-Based | $0.68 \pm 0.44$ | $1.85 \pm 1.12$ | $\mathbf{2.30 \pm 1.24}$ | $\mathbf{1.61 \pm 0.68}$ | $\mathbf{1.18 \pm 0.30}$ |
| Data-Based | $\mathbf{0.65 \pm 0.30}$ | $\mathbf{1.65 \pm 0.77}$ | $2.60 \pm 1.30$ | $1.63 \pm 0.55$ | $1.24 \pm 0.22$ |

Table S2: Robustness to template initialization on MMWHS-4. We compare the original heart template to a strongly sheared variant that the alignment module cannot undo. *Per-class Chamfer distance ($\times 10^{-3}$) degrades only slightly.*

| Template | Intersection Classes | | | | Other Classes |
|---|---|---|---|---|---|
| | LV ∩ LA | LV ∩ RV | RV ∩ RA | Average | Average |
| Original | 0.37 | 1.75 | 1.00 | 1.04 | 0.98 |
| Sheared | 0.40 | 2.23 | 0.75 | 1.13 | 1.09 |

terms of reconstruction loss, using the data template underperforms only very slightly as shown in Tab. S1, but the reconstructed meshes are significantly less regular as shown in Fig. S1.

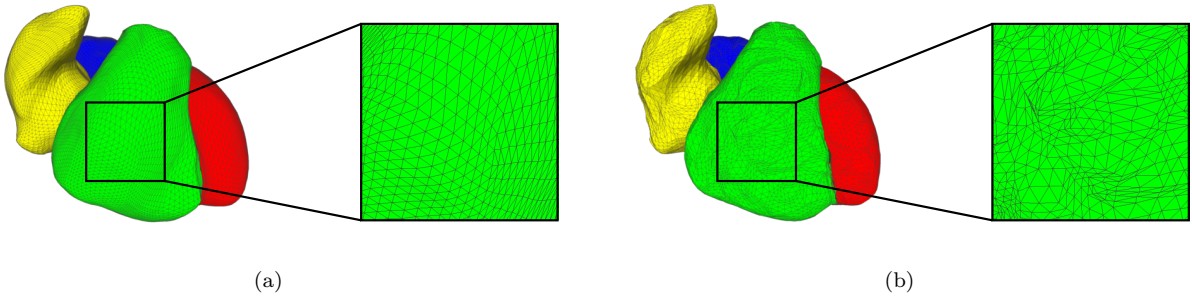

(a)                (b)

Figure S1: **Regular vs Data-Based Template.** (a) Reconstruction using the hand-crafted template of Fig. 2c. (b). Reconstruction using a data-based template. While it yields similar numerical reconstruction performance is similar, the triangles in the data-based version are much less regular and the resulting meshes less suitable for downstream applications, such as simulation.

## C.2   Robustness to Template Variation

To probe how sensitive the method is to template initialization quality, we deliberately distort the heart template before training. A naive "bad template" obtained by translating, rotating, or rescaling the original isotropically will have almost no effect, as our alignment module removes any such global pose and scale offsets before deformation.

We therefore apply a strong shear about the centroid ($xy = 0.6$, $xz = 0.4$, $zy = 0.3$, det $= 1$) and rescale the result isotropically into the original bounding box, as shown in Fig. S2. This skews the proportions of the template incorrectly. Everything else, including the use of alignment at training time, is left unchanged. We re-train the model on a single MMWHS-4 split and compare against the unmodified baseline in Tab. S2.

The impact on reconstruction quality is small, but present: per-class Chamfer distance worsens by roughly 10% on average. This shows that our approach can produce reasonable estimates even with unusually shaped templates, but best performance is obtained with a good template.

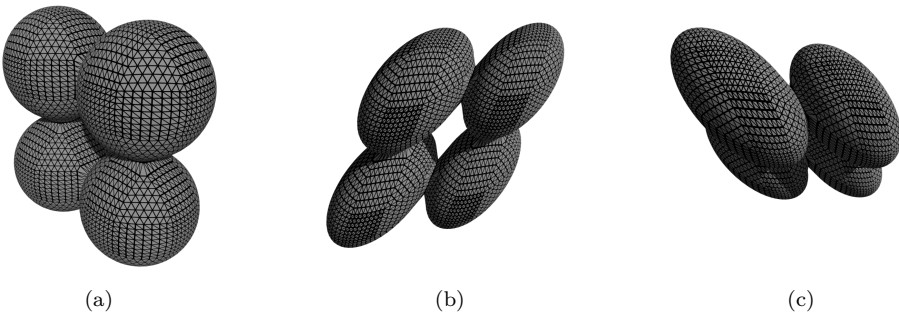

(a)          (b)          (c)

Figure S2: **Original (a) vs sheared (b), (c) heart template.**

### C.3  Freezing the segmentation network

With a frozen segmentation network (backbone), the mesh reconstruction performance is significantly reduced. On MMWHS-4, the avg. Chamfer distance ($\times 10^{-3}$) is $1.25 \pm 0.47$ with a refined backbone, vs. $1.38 \pm 0.49$ with a frozen one. We believe this is due to the pre-trained segmentation features lacking specific information for 3D surface deformation.

### C.4  Geometric Regularization Weight

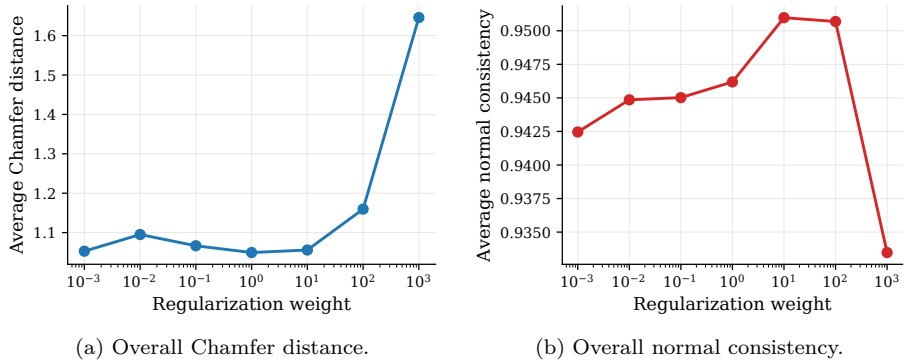

(a) Overall Chamfer distance.     (b) Overall normal consistency.

Figure S3: **Effect of the geometric regularization weight.** We scale the combined weight of the geometric regularization loss terms and report average Chamfer distance and normal consistency on a single split of MMWHS-4.

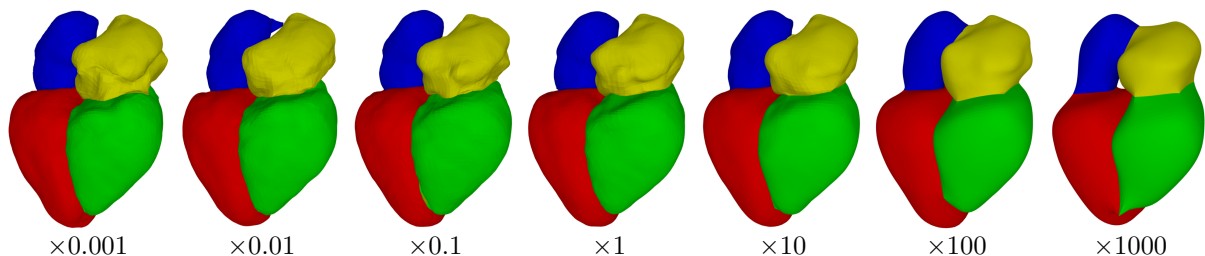

Figure S4: **Reconstructed meshes across regularization weights.** The same sample reconstructed with the geometric regularization weight scaled from $\times 0.001$ (left) to $\times 1000$ (right), with our default at $\times 1$. With too little regularization the surface is rough and irregular, while excessive regularization over-smooths the mesh and collapses fine structure, sacrificing reconstruction accuracy.

The geometric regularization loss terms of Sec. A trade off reconstruction fidelity against mesh quality. To quantify their effect, we scale their combined weight by factors from $\times 0.001$ to $\times 1000$ relative to our default ($\times 1$) and evaluate on a single split of MMWHS-4. Fig. S3 reports the overall average Chamfer distance and normal consistency as a function of this factor. Both metrics are stable across roughly three orders of magnitude around the default, indicating that our method is not very sensitive to the precise value of the regularization weight. Performance only degrades clearly once the regularization weight grows or shrinks significantly enough to dominate the reconstruction loss ($\times 100$ and beyond), where the Chamfer distance rises and normal consistency drops sharply.

Fig. S4 shows the same reconstructed sample across all seven settings. With too little regularization the surface is prone to containing artifacts and protrusions, whereas excessive regularization over-smooths the mesh. The default setting offers a good compromise between surface smoothness and reconstruction fidelity, but arguably nearby regularization weights up to an order of magnitude produce similar results.

