# OpenReview forum: "PrIntMesh: Precise Intersection Surfaces for 3D Organ Mesh Reconstruction"
_TMLR — Decision pending for TMLR_

### Review · Reviewer_pyTP · 2026-04-28

**Summary Of Contributions:**

This paper introduces PrIntMesh, a voxel-to-mesh approach for reconstructing organs while enforcing topological consistency, such as smooth and artifact-free surfaces. The method starts from a predefined template, which can be manually designed or automatically generated, and deforms it to match the input voxel grid. A 3D U-Net is first used to extract voxel-wise segmentation and feature maps, which are then used to align the template to the target anatomy. The aligned template is iteratively refined through a mesh deformation network that updates vertex positions based on sampled voxel features. To ensure accurate and consistent reconstruction, the method introduces an interface-aware supervision scheme that explicitly models both individual structures and their shared boundaries. The model is trained using a combination of voxel-based losses (cross-entropy and Dice) and mesh-based losses that encourage geometric accuracy and smoothness.

The authors introduce an interesting approach to organ reconstruction using a template-based framework, incorporating ideas such as interface decomposition to explicitly model shared boundaries between anatomical structures. This is a meaningful addition, as it allows the method to better handle interactions between components that are often treated independently in prior work. The approach addresses known limitations of voxel-based and independent mesh-based methods, particularly in enforcing topological consistency. The results show that the method is capable of producing smooth, artifact-free, and gap-free reconstructions, which is important for downstream applications.

However, there are several aspects that, if addressed, would strengthen the paper. First, there is limited ablation analysis demonstrating the contribution of individual components, such as the interface decomposition and the various regularization terms. A more detailed breakdown would help clarify which parts of the method are most impactful. It would also be useful to better understand the role of the template itself. For example, how sensitive is the method to the choice or design of the template? If the template does not closely reflect the target anatomy, does performance degrade? Exploring variations, such as modifying the placement or connectivity of the underlying primitives, would provide insight into the robustness of the approach. Additionally, the novelty appears somewhat incremental, as the method builds heavily on prior work such as Voxel2Mesh, with the primary differences being the use of a unified template and interface-aware supervision. While these are meaningful additions, the overall framework remains similar. Finally, the approach consists of multiple stages, including voxel-based feature extraction, alignment, iterative mesh deformation, and subdivision, which may introduce additional computational overhead. A clearer comparison of runtime and efficiency relative to the baselines would help better assess the practical trade-offs of the method.

**Audience:**

Yes

**Audience Explanation:**

This paper would be of interest to members of the TMLR audience, particularly those working in 3D vision, medical imaging, and structured geometric learning. The proposed method introduces a unified template and interface-aware supervision, which are meaningful ideas for enforcing structural consistency in mesh reconstruction. Ihas practical relevance and may be of interest to researchers working on geometry-aware learning and reconstruction.

**Claims And Evidence:**

Yes

**Claims Explanation:**

The authors provide both qualitative and quantitative evidence demonstrating that their approach produces smooth, gap-free, and topologically consistent reconstructions. However, the overall strength of the evidence could be improved with more detailed analysis, such as ablation studies isolating key components of the method.

**Requested Changes:**

The authors should include additional analysis to better isolate the contribution of individual components of the method, such as the interface decomposition, alignment step, and regularization terms. This would be critical to understanding the effectiveness of the proposed design and to support the main claims. Additionally, a comparison of computational efficiency and runtime relative to the baselines would be important for assessing the practical applicability of the approach. To help strengthen the paper, it would be useful to include qualitative visualizations of the DeepCSR method, which appears to be a strong competing approach. This would provide clearer insight into the differences in reconstruction quality between methods.

---

> ### Author Response · Authors · 2026-06-19
> **Addressing Requested Changes**
>
> We thank the reviewer for their in-depth review of our method. Their requests for ablations isolating each component led to several new experiments that further clarified our contributions. We address the requested changes below:
>
> > **Q1:** ... additional analysis to better isolate the contribution of interface decomposition.
>
> We have added an experiment in our ablation studies, as "Interface Supervision" (Section 4.5, Table 7). In the experiment, we remove the explicit supervision on shared areas, and instead only supervise the main classes. Since the shared interfaces get conflicting supervision signals from multiple main classes, an arbitrary patch ends up at the intersection, and the performance for the shared interfaces degrades significantly. The overall performance also degrades slightly, as we end up removing the interface information from the supervision.
>
> Per-class Chamfer distance (×10⁻³) on a single MMWHS-4 split:
>
> | Template | LV∩LA | LV∩RV | RV∩RA | Intersection Avg | Other Avg |
> |---|---|---|---|---|---|
> | Shared-surface (ours) | 0.31 | 2.59 | 0.75 | **1.22** | 1.05 |
> | Main-surface only | 0.57 | 22.07 | 2.06 | **8.23** | 1.09 |
>
> > **Q2:** ... additional analysis to better isolate the contribution of alignment step.
>
> We have added a new ablation that disables the alignment module entirely (Section 4.5, Table 8). The deformation network is then asked to recover the full rigid and scale displacement on its own, which it does not manage well: average per-class Chamfer distance rises by 19\%, with the error concentrated on the shared-surface intersection classes (+26\% on the intersection average vs. +13\% on the standalone chambers), and inverted faces appearing on several classes. The alignment step is therefore an important contributor to both accuracy and output mesh quality.
>
> Per-class Chamfer distance (×10⁻³) on a single MMWHS-4 split:
>
> | Variant | LV∩LA | LV∩RV | RV∩RA | Intersection Avg | Other Avg |
> |---|---|---|---|---|---|
> | With alignment | 0.37 | 1.75 | 1.00 | **1.04** | 0.98 |
> | Without alignment | 0.36 | 2.23 | 1.35 | **1.31** | 1.11 |
>
> > **Q3:** ... additional analysis to better isolate the contribution of regularization terms.
>
> While these terms are commonly used in similar 3D deformation models, we agree that seeing their exact effect would leave a stronger impression about their usefulness. We designed a new ablation where we take all the regularization terms together and scale their combined strength by factors ranging from ×0.001 to ×1000 relative to our default (×1), evaluating on a single MMWHS-4 split. We have added this ablation to the paper (Appendix C.4), with the quantitative plots in Figure S2 and the qualitative mesh visualizations in Figure S3.
>
> The overall reconstruction quality is stable across several orders of magnitude around the default, and only degrades once the regularization grows strong enough to dominate the reconstruction loss (×100 and beyond). The qualitative meshes (Figure S3) show that too little regularization leaves surface artifacts and protrusions, while too much over-smooths the mesh.
>
> Overall average Chamfer distance (×10⁻³, lower is better) and normal consistency (higher is better) on a single MMWHS-4 split:
>
> | Reg. weight | Chamfer | Normal consistency |
> |---|---|---|
> | ×0.001 | 1.05 | 0.942 |
> | ×0.01 | 1.10 | 0.945 |
> | ×0.1 | 1.07 | 0.945 |
> | ×1 (default) | **1.05** | 0.946 |
> | ×10 | 1.06 | **0.951** |
> | ×100 | 1.16 | **0.951** |
> | ×1000 | 1.65 | 0.933 |

---

> > ### Author Response · Authors · 2026-06-19
> > **Addressing Requested Changes #2**
> >
> > > **Q4:** Additionally, a comparison of computational efficiency and runtime relative to the baselines.
> >
> > We have added this comparison to the main paper (Section 4.6, Table 9). We time every method in the same default configuration used for its reported accuracy, so the timing and accuracy numbers stay consistent. On MMWHS-4, our runtimes are comparable to MeshDeformNet, the most similar mesh-based baseline: our per-step training time (0.51s) is in the same range, while our per-sample inference time (0.11s) is in fact the fastest among all methods. Relative to a pure voxel segmentation network, we only add a graph convolution network on top, which has a small impact at inference time, as is also the case for other graph-based methods like MeshDeformNet and DeepCSR. nnU-Net has the fastest training, since it is only a segmentation network, but the slowest inference, as its default applies test-time augmentation, averaging 8 mirrored predictions. The remaining differences come down to implementation details, e.g. MedTet's high-resolution grid is expensive to supervise, whereas DeepCSR avoids mesh extraction during training.
> >
> > Per-step training and per-sample inference time (mean ± std, seconds):
> >
> > | Method | Training (s/step) | Inference (s/sample) |
> > |---|---|---|
> > | nnU-Net | 0.182 ± 0.003 | 0.330 ± 0.001 |
> > | MeshDeformNet | 0.650 ± 0.010 | 0.139 ± 0.004 |
> > | DeepCSR | 0.282 ± 0.035 | 0.243 ± 0.001 |
> > | MedTet | 1.778 ± 0.155 | 0.585 ± 0.037 |
> > | **PrIntMesh (Ours)** | **0.510 ± 0.014** | **0.113 ± 0.005** |
> >
> > > **Q5:** To help strengthen the paper, it would be useful to include qualitative visualizations of the DeepCSR method, which appears to be a strong competing approach.
> >
> > We agree with this assessment and have rebuilt Figure 4 to include a visualization for DeepCSR as well.

---

### Review · Reviewer_pBsY · 2026-06-01

**Summary Of Contributions:**

PrIntMesh proposes a template-based framework for reconstructing multi-part organs (heart, hippocampus, lungs) as unified mesh systems rather than independent substructures. The method starts from a topologically-correct connected template (built from rhombicuboctahedra), deforms it with a two-stream network (3D U-Net + graph convolutional mesh decoder), and enforces interface preservation through a decomposition of surfaces into exclusive and shared regions with dedicated Chamfer supervision.

Strengths

- The paper clearly articulates why independently reconstructing organ substructures is insufficient — gaps, floaters, interpenetrations all break downstream tasks like CFD simulation and surgical planning. Using rhombicuboctahedra as primitives that naturally share square faces is a clever construction. The one-time manual effort is minimal, and the automated template generation from marching cubes is a good fallback.

- The formalization of Interface decomposition into exclusive surfaces and interface surfaces, with equal-weight Chamfer supervision per region, is the paper's core technical contribution and is well-conceived. It prevents shared boundaries from being overwhelmed by larger structures. It also provides strong topology guarantees. Zero intersection volume and zero unwanted gaps, versus non-zero for all baselines, is a compelling result. The topology is guaranteed by construction, not by soft loss.

- The 2-3x Chamfer improvement over nnU-Net in the low-data regime (Figure 6) is practically significant for clinical settings with limited annotations. The paper also demonstrates broader applicability on three anatomically distinct organs (4-chamber heart, 2-part hippocampus, 5-lobe lungs) shows generality.

Weaknesses

- Very small evaluation dataset for the heart. MM-WHS has only 20 training CTs with 5-fold cross-validation. With such small numbers and high variance. it's hard to draw statistically robust conclusions. No significance tests are reported.

- Limited baselines for topology-aware methods. The comparisons are against nnU-Net, MeshDeformNet (2021), DeepCSR (2021), and MedTet (2024). The paper acknowledges these are "a few years old but not yet superseded." However, recent topology-loss methods (Gupta et al. 2022, Xu et al. 2025) are only discussed, not compared against experimentally. These would be the most meaningful competitors since they also target topological correctness.

- The paper claims the method generalizes by "simply changing the template." But the template design requires knowing the organ's topology a priori. For organs with variable topology (e.g., vascular trees, tumors with variable connectivity), the approach may not apply. This limitation is underexplored.

- Dice scores are essentially equivalent to nnU-Net (Table 2). The improvements are concentrated at boundaries, which the paper acknowledges. A reviewer could argue: if you care about topology, add a post-processing step to nnU-Net (which the paper dismisses but doesn't experimentally ablate against).

**Audience:**

Yes

**Audience Explanation:**

Yes. The paper addresses a real gap at the intersection of geometric deep learning and medical imaging that is relevant to multiple TMLR subcommunities:

**Claims And Evidence:**

Yes

**Claims Explanation:**

The topological correctness and data efficiency claims are well-supported. The accuracy claims are supported but weakened by small sample sizes and lack of statistical testing. Overall, the core technical claims are adequately supported, but the paper oversells its practical impact without corresponding validation.

**Requested Changes:**

Please explain the following questions to make the paper more complete:
- How sensitive is the method to template initialization quality? What happens with a poor initial template (e.g., wrong proportions)?
- Can you provide training/inference times relative to baselines?
- What is the failure mode? When does the deformation network fail to converge to the target anatomy?

---

> ### Author Response · Authors · 2026-06-19
> **Addressing Requested Changes**
>
> We thank the reviewer for their thorough and detailed feedback on our submission. Their questions on failure modes and statistical tests pushed us to strengthen several parts of the paper. We address their stated concerns below:
>
> > **Q1:** How sensitive is the method to template initialization quality? What happens with a poor initial template (e.g., wrong proportions)?
>
> Generic, high-level problems in the template such as simple translations, scaling and rotations are taken care of during the alignment step. Thus, our training is robust to templates that have been scaled isotropically, rotated or moved.
>
> The exact structure of the template has little effect on overall reconstruction performance: as shown in our ablation study "Data-Based Template Generation" (Appendix C.1, Table S1), simply using a held-out sample from the dataset as an initial template yields almost the same reconstruction performance.
>
> As a more extreme test, we applied an arbitrary shear to the heart template and recorded the result as another ablation study (Appendix C.2, Table S2). The reconstruction quality only degrades slightly even under this strong distortion (per-class Chamfer distance, ×10⁻³, on a single MMWHS-4 split):
>
> | Template | LV∩LA | LV∩RV | RV∩RA | Intersection Avg | Other Avg |
> |---|---|---|---|---|---|
> | Original | 0.37 | 1.75 | 1.00 | 1.04 | 0.98 |
> | Sheared | 0.40 | 2.23 | 0.75 | 1.13 | 1.09 |
>
>
> > **Q2:** Can you provide training/inference times relative to baselines?
>
> We have added this comparison to the main paper (Section 4.6, Table 9). We time every method in the same default configuration used for its reported accuracy, so the timing and accuracy numbers stay consistent. On MMWHS-4, our runtimes are comparable to MeshDeformNet, the most similar mesh-based baseline: our per-step training time (0.51s) is in the same range, while our per-sample inference time (0.11s) is in fact the fastest among all methods. Relative to a pure voxel segmentation network, we only add a graph convolution network on top, which has a small impact at inference time, as is also the case for other graph-based methods like MeshDeformNet and DeepCSR. nnU-Net has the fastest training, since it is only a segmentation network, but the slowest inference, as its default applies test-time augmentation, averaging 8 mirrored predictions. The remaining differences come down to implementation details, e.g. MedTet's high-resolution grid is expensive to supervise, whereas DeepCSR avoids mesh extraction during training.
>
> Per-step training and per-sample inference time (mean ± std, seconds):
>
> | Method | Training (s/step) | Inference (s/sample) |
> |---|---|---|
> | nnU-Net | 0.182 ± 0.003 | 0.330 ± 0.001 |
> | MeshDeformNet | 0.650 ± 0.010 | 0.139 ± 0.004 |
> | DeepCSR | 0.282 ± 0.035 | 0.243 ± 0.001 |
> | MedTet | 1.778 ± 0.155 | 0.585 ± 0.037 |
> | **PrIntMesh (Ours)** | **0.510 ± 0.014** | **0.113 ± 0.005** |
>
> > **Q3:** What is the failure mode? When does the deformation network fail to converge to the target anatomy?
>
> We have added a Limitations section to the main paper (Section 5) that collects the principal failure modes. In short:
>
> * **Alignment failure from segmentation errors.** Template alignment depends on the segmentation network, since the similarity transform is estimated from the predicted class centroids. A poor segmentation degrades the alignment and leaves the deformation network with a bad starting point. Jointly training the segmentation network with the mesh decoder reduces this dependency but does not remove it.
> * **Topology mismatch.** Because we enforce topology by construction, the network cannot adapt when a patient's anatomy violates the template's connectivity assumptions, e.g. anomalies where parts that are normally connected are not. The network still produces an output matching the template's topology, but not the patient's.

---

> > ### Author Response · Authors · 2026-06-19
> > **Addressing Requested Changes #2**
> >
> > > **Q4:** Weaknesses, such as no significance tests for MMWHS-4, Gupta et al. 2022 & Xu et al. 2025 not being compared to, modeling variable topology, and adding postprocessing to nnU-Net.
> >
> > 1. We thank the reviewer for highlighting the need for statistical testing. On reflection, we agree that reporting our statistics at the fold level was not the best choice: with only $n=5$ folds, whose training sets overlap heavily, fold-level statistics carry little power and cannot support a sound significance test.
> >
> >     We have therefore moved the analysis to the level of the individual hearts. We pool the out-of-sample predictions from every fold, so that each of the $20$ annotated scans is held out exactly once, and report the mean and standard deviation over these $20$ predictions in Table 1, rather than over the folds. Because the folds are of equal size, this leaves the reported means unchanged and only affects the standard deviations. For significance, since Chamfer errors are skewed and long-tailed and therefore not normally distributed, we use a paired Wilcoxon signed-rank test on these per-heart predictions, comparing our model against each baseline one by one. The results support our conclusion: for most baselines we obtain a clear improvement in intersection performance, while remaining similar in overall reconstruction. The only baseline where the difference is not significant is nnU-Net, which is understandable, as its voxel segmentation prevents intersections and is already roughly correct; our comparative advantage is the guarantee of a well-connected and smooth intersection surface.
> >
> >     We have added these significance tests to the experiments section of the main paper (Section 4.4, Table 4). The per-class Wilcoxon p-values are reported below (bold marks $p < 0.05$, values below 0.001 are reported as <0.001):
> >
> >     | Baseline | LV∩LA | LV∩RV | RV∩RA | Intersection Avg | Other Avg |
> >     |---|---|---|---|---|---|
> >     | MedTet | **0.040** | **0.017** | 0.430 | 0.064 | 0.869 |
> >     | DeepCSR | **<0.001** | **0.001** | 0.159 | **<0.001** | 0.812 |
> >     | MeshDeformNet | 0.064 | **0.003** | 0.070 | **0.004** | **0.006** |
> >     | nnU-Net | 0.114 | 0.596 | 0.261 | 0.216 | 0.927 |
> >
> >
> >
> > 2. We thank the reviewer for mentioning these topology-relevant baselines.
> >
> >     The main contribution of  Gupta et al. (2022)  is a loss function applicable to any segmentation method, and is applicable to the nnU-Net baselines. However, their loss relies on two interactions: k-voxel wide separation between classes (exclusion), and full containment of one class inside the other. Then, these losses are calculated with masks. By definition, voxel-based methods already cannot have intersections due to each voxel being occupied by one class, and we want our classes to touch, not be separated. There's no way to enforce voxels to "touch around a region" or "not have disconnected holes in the surface between regions", which is what our approach does. As such, we believe this loss offers no improvement for the particular topological issues that we address.
> >
> >     While Xu et al. (2025) is a notable recent work about topology in medical imaging, its methodology is specifically tailored for instance separation in histopathology (nuclei segmentation). The underlying assumptions and spatial priors of this approach do not straightforwardly translate to the reconstruction of macroscopic multi-part organ meshes. Therefore, a direct comparison was not technically feasible for our specific task.
> >
> > 3. As discussed in our new Limitations section (Section 5) and in our answer to Q3, our current formulation relies on a strict topological prior by design. For organs with consistent connectivity, this is what guarantees the intersection-free meshes we reconstruct. It does not directly generalize to structures with highly variable, patient-specific connectivity. We leave supporting these cases, e.g. via multi-template selection or topology-adaptive deformation, for future work.
> >
> > 4. We thank the reviewer for this point. To clarify our setup, we do use the default simple post-processing for nnU-Net by keeping only the largest connected component for each class, which removes spurious disconnected fragments ("floaters"). While it would be possible to design a custom post-processing pipeline for nnU-Net to address more topological errors (e.g., multi-label hole-filling, and iterative collision resolution), this would be computationally expensive and prone to edge case failures. In contrast, our approach guarantees this correctness by construction, which we believe is more robust than relying on heuristic multi-stage post-processing pipelines.

---

### Review · Reviewer_6v4k · 2026-06-05

**Summary Of Contributions:**

This paper studies the problem of reconstructing 3D meshes of human organs from medical images. The idea is to design a neural network that extracts features from images, and then use these features to guide mesh deformation from a template that has the same topology as the target organ. The authors also proposed loss functions to ensure the results maintain geometry integrity. Experimental results show that the proposed method outperform existing methods in terms of reconstruction correctness.

**Audience:**

Yes

**Audience Explanation:**

This paper discuss learning based method to solve the 3D organ reconstruction problem, which should be interesting to some of TMLR's audience working in this area.

**Broader Impact Concerns:**

I have no broader impact concern.

**Claims And Evidence:**

Yes

**Claims Explanation:**

The paper demonstrates thorough experimental evidences with different organs comparing to multiple existing methods to show the improvements brought by the proposed technique as claimed by the paper.

**Requested Changes:**

The paper is overall clearly written and well structured. My only concern is that the paper seems to be missing some details on how exactly the mesh is deformed and how the subdividing works. The authors are encouraged to revise the manuscript to include the following details:
1. What does the mesh decoder predict to deform the template mesh? How is the data structure designed to handle the 3D data flow? Is there any limitation in the scale (e.g. number of triangles in the mesh) of the mesh that the model can process due to for example memory constraints?
2. How does the mesh subdivision work? Is the subdivision regular (i.e. one triangle divided to a fixed number of sub-triangles)? Why is it designed this way?
Providing these details will make the paper more self-contained.

---

> ### Author Response · Authors · 2026-06-19
> **Addressing Requested Changes**
>
> We thank the reviewer for their thoughtful review, and are glad that they found our approach interesting. Their questions gave us the opportunity to make our design choices explicit in the paper. We address their requests below:
>
> > **Q1:** What does the mesh decoder predict to deform the template mesh? Data flow? Scale limitation?
>
> We thank the reviewer for this question and added the additional details below to the architecture part of the appendix (Appendix B.1).
>
> Each stage of the mesh decoder predicts only a per-vertex 3D deformation, essentially of shape (V, 3) where the input has V vertices. V starts as the number of vertices in the template, and later increases with each subdivision.
>
> Regarding the data flow, the full mesh has three components. Two are standard in every mesh: a (V, 3) floating-point vertex array storing vertex coordinates, and an (F, 3) integer triangle/face array pointing to the three composing points of each triangle. We add another integer array of shape (F,) that stores face classes, the class of each triangle. These stay fixed during the deformation, barring subdivision where each new triangle inherits the class of the triangle of their parent. Shared interfaces use a paired encoding: given the two adjoining classes c₁ and c₂, we encode them as c = K · c₁ + c₂ with K = 1000. This encoding is only important for evaluation when trying to extract a "full" part, during training the shared interfaces are treated the same as main classes: individually.
>
> We keep our mesh resolutions close to those of the methods we base ourselves on, which is a good fit for our 128×128×128 voxel input. Beyond this, there is a limit on the number of triangles due to memory, but it is high. For our hearts experiment, we use a single subdivision and have an output mesh with around 43K triangles. During our tests, we checked that it is possible to run training on an 80GB A100 GPU with up to 2.7 million output triangles. Inference can go even higher. In practice, we find ourselves more limited by compute during training than by memory. With our original size, one iteration takes about 0.4 seconds, while it takes almost 20 seconds in the 2.7M triangle setting.
>
> > **Q2:** How does the mesh subdivision work?
>
> We use regular mesh subdivision (see [PyTorch3D's SubdivideMeshes](https://pytorch3d.readthedocs.io/en/latest/modules/ops.html#pytorch3d.ops.SubdivideMeshes)), where one triangle is divided into four new triangles by adding new edges between the midpoints of each existing edge. We decided to use this simple approach rather than the adaptive unpooling of the original  Voxel2Mesh (Wickramasinghe et al., 2020) for two reasons.
>
> First, we found that caching and re-using the same triangle topology at every training step sped up training, at the sole cost of having more vertices in the final output and without any significant impact on performance.
>
> Second, and perhaps more importantly, the adaptive unpooling method (Wickramasinghe et al., 2020) works by first dropping a fixed ratio of vertices from the denser regions of the deformed shape, and then retriangulating the original template with a convex hull algorithm. This works in the case where the template is a convex shape like a simple sphere, which is the case in most previous methods, but not for our case with unified multi-part templates. If we modify the template, we would not be able to recover a new one via convex hull, and applying a similar approach would need a significant redesign.
>
> We have also added a discussion on these details to the appendix (Appendix B.1).

---

### Author Response · Authors · 2026-06-19
**Paper Revision after Reviews**

We are grateful to the reviewers for the time and effort they spent on evaluating our paper. We did our best to address their concerns and requested changes, as detailed below. We temporarily colored our changes in the revised version of the paper in blue to make them stand out.

Beyond these, we also fixed several small grammar mistakes and incorrect Figure/Table references, along with a single misreported numeric value. The standard deviation of the number of holes for nnU-Net in Table 4 was mistyped as 0.85 ± 0.02; the correct value under the original calculation was 0.85 ± 0.52. With the revised per-heart standard-deviation calculation (see our Q4 answer 1 to reviewer pBsY below), this became 0.85 ± 1.14 in the updated table.